# Geography as non-genetic modulation factor of chicken cecal microbiota

**Natalia Pin Viso[1,2], Enzo Redondo[2,3], Juan María Díaz Carrasco[2,3], Leandro Redondo[2,3], Julia Sabio y. Garcia[1], Mariano Fernández Miyakawa[2,3], Marisa Diana Farber[1,2] ***

**1** Instituto de Agrobiotecnología y Biología Molecular, IABiMo, INTA-CONICET, Calle Las Cabañas y Los Reseros s/n, Castelar, Buenos Aires, Argentina, **2** Consejo Nacional de Investigaciones Científicas y Técnicas, Ciudad Autónoma de Buenos Aires, Buenos Aires, Argentina, **3** Instituto de Patobiología Veterinaria, IPVet, INTA-CONICET, Calle Las Cabañas y Los Reseros s/n, Castelar, Buenos Aires, Argentina

* farber.marisa@inta.gob.ar

**Data Availability Statement:** All sequence data were deposited in the NCBI Sequence Read Archive database under the BioProject accession number PRJNA579062. Publicly available from now on.

## Abstract

The gastrointestinal tract of chickens harbors a highly diverse microbiota contributing not only to nutrition, but also to the physiological development of the gastrointestinal tract. Microbiota composition depends on many factors such as the portion of the intestine as well as the diet, age, genotype, or geographical origin of birds. The aim of the present study was to demonstrate the influence of the geographical location over the cecal microbiota from broilers. We used metabarcoding sequencing datasets of the 16S rRNA gene publicly available to compare the composition of the Argentine microbiota against the microbiota of broilers from another seven countries (Germany, Australia, Croatia, Slovenia, United States of America, Hungary, and Malaysia). Geographical location played a dominant role in shaping chicken gut microbiota (Adonis R2 = 0.6325, $P$ = 0.001; Mantel statistic r = 0.1524, $P$ = 4e-04) over any other evaluated factor. The geographical origin particularly affected the relative abundance of the families Bacteroidaceae, Lactobacillaceae, Lachnospiraceae, Ruminococcaceae, and Clostridiaceae. Because of the evident divergence of microbiota among countries we coined the term "local microbiota" as convergent feature that conflates non-genetic factors, in the perspective of human-environmental geography. Local microbiota should be taken into consideration as a native overall threshold value for further appraisals when testing the production performance and performing correlation analysis of gut microbiota modulation against different kind of diet and/or management approaches. In this regard, we described the Argentine poultry cecal microbiota by means of samples both from experimental trials and commercial farms. Likewise, we were able to identify a core microbiota composed of 65 operational taxonomic units assigned to seven phyla and 38 families, with the four most abundant taxa belonging to *Bacteroides* genus, Rikenellaceae family, Clostridiales order, and Ruminococcaceae family.

## Introduction

The chicken gastrointestinal tract (GIT) harbors a very diverse microbiota, dominated by Bacteria, that influences health and growth performance of chicken. A healthy microbiota

**Funding:** This work was supported by the Consejo Nacional de Investigaciones Científicas y Técnicas (CONICET) and National Institute of Agricultural Technology (INTA). N.P.V. has a fellowship from CONICET; L.R, J.M.D.C, M.F.M. and M.D.F. are members of research career of CONICET and INTA, Argentina. INTA- PNBIO 1131043/PNSA I106 to M. D.F and PNSA 1115056/I104 to M.F.M. The funders had no role in study design, data collection and analysis, decision to publish, or preparation of the manuscript.

**Competing interests:** The authors have declared that no competing interests exist.

provides nutrients to the host and promotes competitive exclusion [1]. The composition of the GIT microbiota differs according to diet, age and genotype of hosts as well as the portion of the intestinal tract, among others [2]). The poultry industry has adopted the use of dietary additives because of their anti-microbial and/or growth-promoting effects. To date, many additives are available, including antibiotics administered in sub-therapeutic doses, prebiotic, probiotics, organic acids and plant extracts [3]).

In the last few years, researchers have been paying closer attention to the influence of the geographical origin on microbiota composition. Indeed, in humans, geography can explain part of the observed variability on intestinal bacterial communities [4, 5]. In birds, environmental parameters seem to be the most important factors shaping host-associated microbiota [6, 7]. For example, fecal microbial composition of egg laying hens and poultry vary between samples from different geographical origins of Europe, such as Slovenia, Hungary, Croatia, and Czech Republic [8]. More recently, by comparing samples from five high altitude regions of China, Zhou *et al.* [9] determined that the cecal composition of the intestinal microbiota of Tibetan chickens is altered by their origin.

With all this in mind, we hypothesized that the geographical location is a key modulator factor of cecal microbiota. In this regard, we considered geography in terms of the "human-environment" theoretical framework [10], overcoming the physical and a human geography divide. Location, as an idealized geographical space [11], would explain flows and interactions between factors like climate, farm management (including health interventions, nutrition, feed and litter management), and socioeconomic and cultural setting. To test this hypothesis, we reanalyzed publicly available data from NCBI and MG-RAST from eight different countries and moved forward using local data for describing the Argentine chicken cecal microbiota.

## Materials and methods

### Data acquisition from public databases

To compare the composition of chicken's GIT microbiota from different geographic locations, short amplicon data from next-generation sequencing experimental trials (ET) were used. Datasets were obtained from MG-RAST and the NCBI Sequence Read Archive. Data sources, corresponding to eight different countries, including Argentina, are reported in Table 1. All downloaded data were re-analyzed using all samples as one large data set, using Quantitative Insights Into Microbial Ecology (QIIME) v1.9.1 software [12].

### Argentine cecal microbiota samples from commercial farms

We selected 10 commercial broiler farms from Argentina, classified according to husbandry practices into conventional poultry (CP) and agroecological farm (AE). Cecal content were collected from 27 samples in total, coming from 9 CP and 1 AE (S1 Table). To reduce inter individual variation each sample is the pool of the cecal content from five animals per pen, after cervical dislocation euthanizing proceeding. All cecum samples were immediately refrigerated on ice and then stored at −80˚C until DNA extraction.

The animal experiments reported in this manuscript were conducted in accordance to protocol number 20/2010 from the Institutional Committee for the care and use of animals-INTA (CICUAE Approved by resolution CICVyA No. 14/07) based on internationally recognized guidelines of "Care and Use of Experimental Animals" as Guide for the Care and Use of Agricultural Animals in Research and Teaching, 3rd edition, 2010. Participation in the study was voluntary.

**Table 1. 16S rRNA gene amplicon data used to compare the chicken GIT microbial composition throughout geographic location.**

| Source/ID | Geographic location | Intestinal portion | Age (days) | Genetic line | Region sequenced | Diet | Extraction kit | Platform | Reference |
|---|---|---|---|---|---|---|---|---|---|
| **NCBI** | | | | | | | | | |
| PRJEB9198 | GER | Cecum | 25 | Ross | complete | SCD, MCP | Qiagen | Roche | [13] |
| SAMN03092832-39 | MAL | Ileum/Cecum | 21/42 | Cobb | V3 | SCD | Qiagen | Illumina | [14] |
| SRP045877 | CRO | Fecal | 21 | Ross/Cobb | V3-V4 | SCD | Qiagen | Roche | [8] |
| SRP045877 | SLO | Fecal | 21 | Ross/Cobb | V3-V4 | SCD | Qiagen | Roche | [8] |
| SRP045877 | HUN | Fecal | 21 | Ross/Cobb | V3-V4 | SCD | Qiagen | Roche | [8] |
| SAMN03161778-871 | USA | Cecum | 42 | Ross/Cobb | V1-V3 | SCD, OA | [15] | Roche | [16] |
| **MG-RAST** | | | | | | | | | |
| 4614960.3 | AUS | Cecum | 25 | Cobb | V1-V3 | SCD | [17] | Roche | [18] |
| **AVAILABLE UNDER REQUEST** | | | | | | | | | |
| -- | ARG-ET1 | Cecum | 26 | Cobb | V3-V4 | SCD, Bac, Tan | Qiagen | Illumina | [19] |
| -- | ARG-ET2 | Cecum | 22 | Cobb | V3-V4 | SCD, Tan | Qiagen | Illumina | Díaz Carrasco Unpublished |

GER: Germany, MAL: Malaysia, CRO: Croatia, SLO: Slovenia, HUN: Hungary, USA: United States, AUS: Australia, ARG-ET1: Argentina-Experimental Trial 1, ARG-ET2: Argentina-Experimental Trial 2. SCD: Standardized commercial diet for feeding broilers. MCP: monocalcium phosphate; OA (Organic acids): formic acid, propionic acid, ammonium formate and medium-chain fatty acids; Bac: subtherapeutic levels of zinc bacitracin; Tan: blend of tannins derived from chestnut and quebracho. Qiagen: QIAmp DNA Stool Mini Kit. Roche: Roche-454. Illumina: Illumina-MiSeq.

## DNA extraction and sequencing

Total genomic DNA was isolated from 300 mg of cecal content using QIAamp DNA Stool Mini Kit (Qiagen, Hilden, Germany) according to the manufacturer's recommendations. DNA concentration and purity were assessed in NanoDrop ND−1000 spectrophotometer (NanoDrop Technologies, DE, USA) and DNA was stored at −20°C until further analysis. The V3-V4 region of bacterial 16S rRNA gene amplification from the total extracted DNA, together with the construction of the 16S gene libraries and high−throughput sequencing using the Illumina MiSeq platform were performed at Macrogen Inc. (Seoul, South Korea). The generated paired-end reads of 300bp were obtained with primers b341F (5'−CCTACG GGNGGCWGCAG−3') and Bakt805R (5'−GACTACHVGGGT ATCTAATCC−3').

All sequence data were deposited in the NCBI Sequence Read Archive database under the BioProject accession number PRJNA579062.

## Microbial community analysis

Microbial community was analyzed by using QIIME v. 1.9.1 with default command parameters, unless specified. An average *phred* quality score threshold higher than 20 was used to filter low quality reads from raw sequence reads. Paired-end reads were joined and potentially chimeric sequences were identified and filtered using UCHIME algorithm [20]. All sequence data were then clustered into operational taxonomic units (OTUs) at 97% similarity against the GreenGenes database version 13.8, using UCLUST algorithm [21]. OTUs with abundance below 0.005% were filtered and the remaining OTUs were normalized using the total-sum

scaling method (which divides the number of sequences per OTU by the total number of sequences in the sample).

For the geographical analysis, due to the broad range of 16S gene regions under analysis, we used the "closed reference" approach. This method discarded reads that failed to match the reference sequences, thus taxonomies came directly from the reference database upon the identity of the reference sequence clustered against.

Microbial diversity was evaluated within samples (alpha diversity) and between samples (beta diversity) using QIIME. Alpha diversity was assessed by richness (Chao1 index and observed OTUs) and community diversity (Shannon and Simpson indexes). The beta diversity in the microbial communities was evaluated on square root transformed OTU abundances; hierarchical clustering was performed on Bray-Curtis dissimilarity by using an average method to grouping the microbiotas in RStudio software with *vegan* package [22]. Finally, UniFrac analysis [23] and unweighted principal coordinate plots (PCoA) were used. For geographical analysis we used the 97% OTUs phylogenetic tree supplied with Greengenes. QIIME scripts used and the intermediate results for this analysis are available from figshare: https://doi.org/10.6084/m9.figshare.c.4993856.

## Statistical analysis of results

To compare the microbial composition of GIT samples from different countries, multiple rarefactions 100 times with averaging count where performing. The output file was further analyzed using Statistical Analysis of Metagenomic Profiles (STAMP) software [24], with ANOVA and Bonferroni correction to identify differentially OTUs abundances. Single sample per country was considered as the experimental unit. To find differences in the alpha diversity indexes, Kruskal-Wallis and Mann-Whitney post-hoc test was realized, corrected by Bonferroni method. For the beta diversity index, the grouping of samples based on each metadata factor, after the PCoA based on unweighted UniFrac distances, was evaluated using non-parametric multivariate ADONIS statistical analysis in QIIME. Additionally, Mantel test, wrapped in RStudio, was performed with 9999 permutations for appraisal the correlation between the unweighted UniFrac distances with geographical ones. We used https://www.geodatos.net/ for geographical coordinates and Haversine distances were calculated using *geosphere* package in R.

For all the statistical analysis, differences at $P < 0.05$ were considered significant.

## Results

### Geographic location shapes the cecal microbiota

By means of comparison analysis of downloaded data from public repositories of eight different countries, we tested the influence of the geographical location on GIT microbiota. The arrangement that arises by the multidimensional scaling analysis (PCoA based on unweighted UniFrac distances of the 16S rRNA gene) revealed distinct groups (Fig 1). Additionally, the fit (63.25% observable variability) of the metadata factors (Table 2) revealed the geographic location as the strongest driver of community structure. On top of that, the Mantel test outcome showed statistically significant correlation between beta diversity and location distance matrices (Mantel statistic r = 0.1524; Significance = 4e-04).

On the other hand, the use of Bray-Curtis dissimilarity metrics in a clustering analysis of microbial communities at the genus level yielded similar results (S1 Fig). Moreover, the relative OTU abundances at the family level also support the community structures associated with geography (Figs 2 and S2 and S2 Table). Notably, the presence of Bacteroidaceae, Lactobacillaceae, Lachnospiraceae, Ruminococcaceae, and Clostridiaceae explained the detected differences (S2 Table).

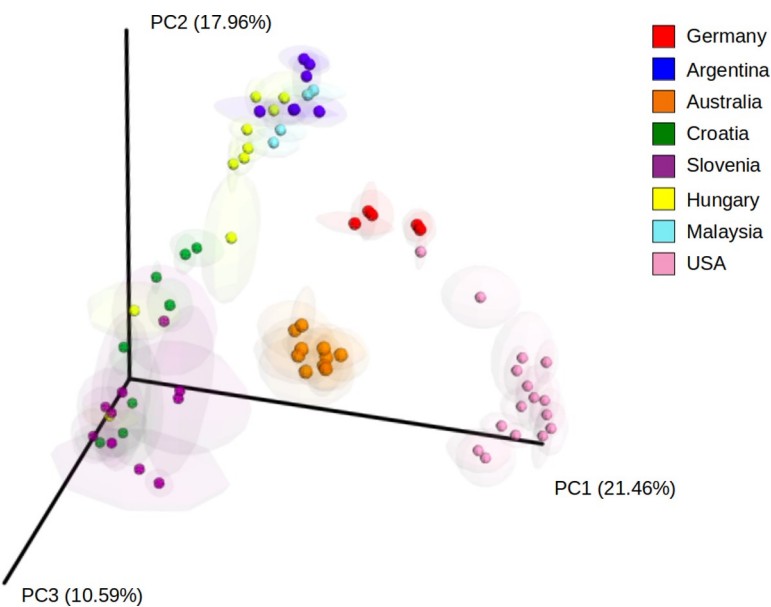

**Fig 1. Principal coordinate analysis (PCoA) of unweighted UniFrac distances.** Chicken GIT microbiota samples from different geographic locations were designated as AUS: Australia, ARG: Argentina, CRO: Croatia, GER: Germany, HUN: Hungary, MAL: Malaysia, SLO: Slovenia and USA: United States.

Additionally, we performed an analysis of the microbiota of each country regarding richness and evenness. In this regard, the community structure of Malaysian microbiota was the most diverse with the higher values of alpha diversity indexes. In contrast, the microbiota from Croatia and USA showed the lower values. Slovenia and Croatia showed no significant statistical differences across Chao1 and Shannon diversity index. These two countries grouped together according to both, PCoA distribution and alpha diversity indexes values. Similarly, alpha and beta values did not support the geographical distance between Hungary, Argentina, and Malaysia (Table 3).

## Characterization of Argentine cecal microbiota

We explored two different categories: chickens reared under experimental trial (ET) and commercial broiler farms (CF). CF included conventional poultry (CP) and one agroecological

**Table 2. Calculated fit of metadata factors to unweighted UniFrac community distances for chicken microbiota using ADONIS.**

| Unweighted UniFrac | R2 |
|---|---|
| Geographic location | 0.63 |
| Region sequenced | 0.35 |
| Diet | 0.36 |
| Extraction Kit | 0.33 |
| Intestinal portion | 0.22 |
| Age (days) | 0.20 |
| Genetic line | 0.17 |
| Platform | 0.11 |

Statistical significance identified for all factor ($P < 0.01$).

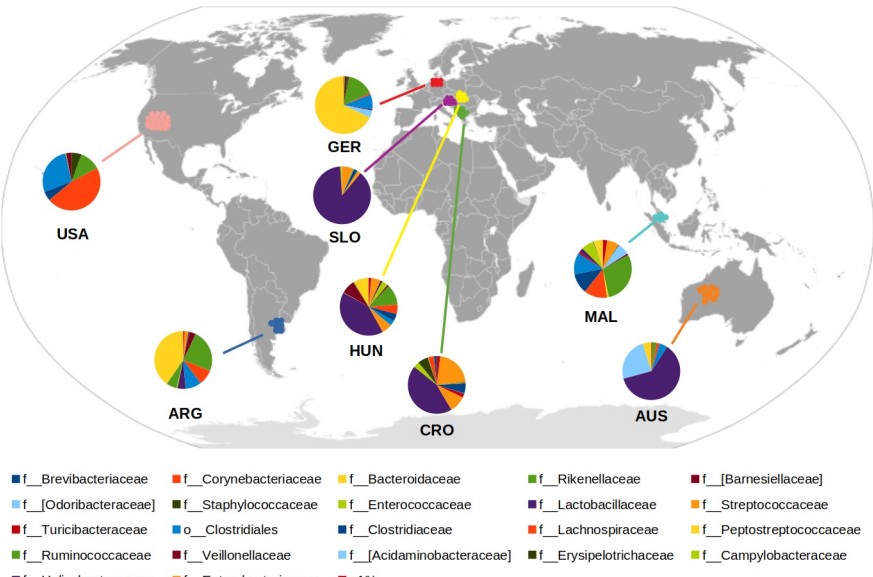

**Fig 2. Geography map of chicken GIT microbiota at the family level.** Different colors are used to indicate each individual taxon according to the country of origin designated as AUS: Australia, ARG: Argentina, CRO: Croatia, GER: Germany, HUN: Hungary, MAL: Malaysia, SLO: Slovenia, and USA: United States. The taxonomic classification: p_phylum, c_class, o_order, and f_family.

farm (AE). The analysis consisted of ET samples involves in the previously analysis, and other 17 samples of the same datasets from different ages (Table 1), and 27 samples obtained from CF (S1 Table). A total of 3032606 quality trimmed sequences and 1172 different OTUs, were obtained with an average number of sequences per sample of 49168, 49881, 69817, and 49297, for ET1, ET2, CP, and AE respectively.

The distribution of the experimental and commercial datasets gathered into different groups according to the multidimensional scaling analysis (PCoA based on unweighted Uni-Frac distances of the 16S rRNA gene). PC1 clearly showed the differences between experimental and commercial datasets, whereas PC2 displayed differences between each experimental trial (ET1 and ET2). Finally, PC3 allowed us to separate the data belonging to AE and CP (Fig 3). In addition, alpha diversity values confirmed two separate groups (ET and CF) according to both richness and diversity indexes (Table 4).

**Table 3. Alpha diversity indexes for chicken samples throughout geographic location.**

| Geographic location | OTUs per sample | Chao1 index | Shannon diversity | Simpson diversity |
|---|---|---|---|---|
| United States | 31.39±5.76a | 37.59±8.78a | 3.09±0.85a | 0.74±0.18ab |
| Croatia | 42.10±26.50a | 55.31±28.10ab | 3.56±0.81ab | 0.84±0.10cd |
| Germany | 44.17±7.33ab | 56.65±8.40abc | 2.90±0.42a | 0.69±0.10a |
| Slovenia | 59.30±10.85bc | 74.20±18.98bc | 2.81±0.61a | 0.72±0.13a |
| Australia | 86.70±15.49bcd | 95.61±20.93cd | 3.25±0.20a | 0.81±0.04abc |
| Hungary | 124.70±43.23cd | 169.41±58.55d | 4.06±0.82bc | 0.85±0.10bcd |
| Argentina | 163.29±32.72d | 189.99±43.41d | 4.57±0.25c | 0.90±0.01d |
| Malaysia | 590.00±35.38e | 621.28±27.78e | 5.92±0.92c | 0.94±0.04d |

Mean±SD are showing. Different letters indicate significant differences among samples according to Kruskal Wallis with Mann-Whitney post-hoc test ($P < 0.05$) and Bonferroni correction.

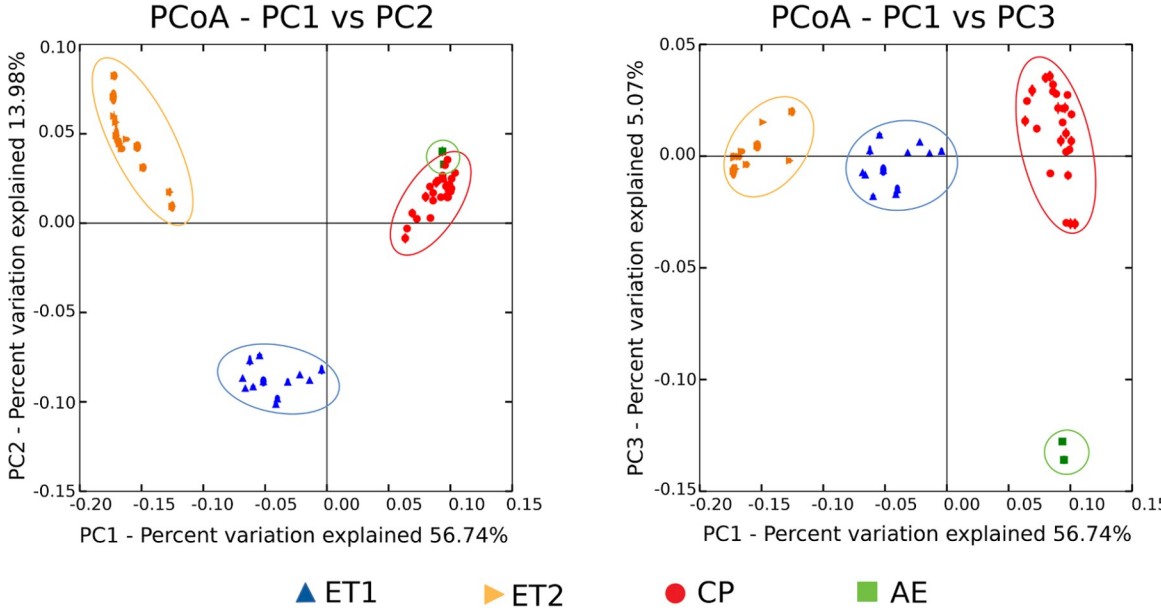

**Fig 3. Principal coordinate analysis (PCoA) of unweighted UniFrac distances.** Chicken cecal microbiota samples from different Argentinian farms were designated as ET1: Experimental Trial 1, ET2: Experimental Trial 2, CP: Conventional Poultry, and AE: Agroecological Farm.

As expected for cecal microbiota, Firmicutes, Bacteroidetes, and Proteobacteria represented the major phyla regarding community structure. For CF, their relative abundances were 45.79%, 46.75%, and 2.52% for CP and of 27.11%, 38.36%, and 8.01% for AE, respectively. The unclassified bacteria ascended to 15.71% in AE samples and 2% in CP. Within the Firmicutes phylum, Clostridia was the dominant class (41.20% for CP and 25.7% for AE), with a prominence of the Clostridiales order. From this order, the prevalent families were Ruminococcaceae and Lachnospiraceae in CP (18.22% and 5.43% respectively) and three families, Ruminococcaceae (4.96%), Lachnospiraceae (4.35%), and Veillonelaceae (7.35%), displayed similar relative abundance in AE samples. Within the Bacteroidetes phylum, Bacteroidales of the Bacteroidea class was a highly abundant order (46.75% for CP and 38.35% for AE). The more representative families from Bacteroidales were Bacteroidaceae (18.56% and 18.28%), Rikenellaceae (15.47% and 1.46%), and Barnesiellaceae (8.39% and 1.27%).

On the other hand, the microbiota of ET was dominated by Firmicutes (50.67%), followed by Bacteroidetes (44.84%), and Proteobacteria (3.41%). Within the Firmicutes phylum, Clostridia was the dominant class (47.23%) among the most abundant members from the Clostridiales order. The most abundant families were Ruminococcaceae and Lachnospiraceae

**Table 4. Alpha diversity indexes for chicken cecal Argentine samples.**

| Sample | OTUs per sample | Chao1 index | Shannon diversity | Simpson diversity | Good's coverage |
|---|---|---|---|---|---|
| ET1 | 328.92±34.20 a | 367.05±34.88 a | 5.27±0.34 a | 0.91±0.02 a | 0.99±0.01 |
| ET2 | 189.58±39.60 b | 218.81±49.63 b | 4.63±0.45 b | 0.90±0.04 a | 0.99±0.01 |
| CP | 571.04±37.11 c | 623.73±43.90 c | 6.56±0.29 c | 0.97±0.01 b | 0.99±0.01 |
| AE | 550.50±17.68 c | 614.39±19.87 c | 6.82±0.11 c | 0.98±0.00 b | 0.99±0.01 |

Mean±SD are showing. Different letters indicate significant differences among samples according to Kruskal Wallis with Mann-Whitney post-hoc test ($P < 0.05$) and Bonferroni correction. ET1: Experimental Trial 1, ET2: Experimental Trial 2, CP: Conventional Poultry, AE: Agroecological Farm

(17.64% and 7.82% respectively). Within the Bacteroidetes phylum, Bacteroidales was the most abundant order of the Bacteroidea class (44.84%). In this case, the most representative families were Bacteroidaceae (33.47%), Rikenellaceae (5.86%), and Barnesiellaceae (5.50%). S3 Table shows the statistical analyses.

Chickens under commercial (CP and AE) or experimental conditions (ET) shared a core microbiota composed of 65 classified OTUs (Fig 4), which were assigned to seven phyla and 38 families. Fifteen of them were above 1% at least in one sample, thus they were considered highly abundant. The shared OTUs were dominated by *Bacteroides* genus, Rikenellaceae family, Clostridiales order, and Ruminococcaceae family. All these taxa showed variations in relative abundance among samples (Table 5).

## Discussion

As in other vertebrates, the gut microbiota composition of birds is influenced by genetic and non-genetic factors. Understanding the contribution of these factors on the microbial community structure is critical to develop a modulation strategy for improving poultry production. Some studies have found that non-genetic factors are more important in structuring the microbiota than the genetics ones [2]. Likewise, our results showed that the geographic location plays a relevant role in shaping the gut microbiota of chickens than any other evaluated factors. This finding is in accordance with previous works studying GIT microbiota modulation in humans and chicken [5, 8, 9].

Particularly, geographic gradient seems to shape microbiota in young European infant (age 6 weeks) [4]. In this age group, the abundance of Bacteroidaceae, Enterobacteriaceae, and Lactobacillaceae remarkably varied according to the location. Indeed all these families presented higher relative abundance in infants from southern European countries. Our results in European chickens showed a similar pattern, with the highest differences found at extreme latitudes. The relative abundances for Lactobacillacea and Enterobacteriaceae were lower in

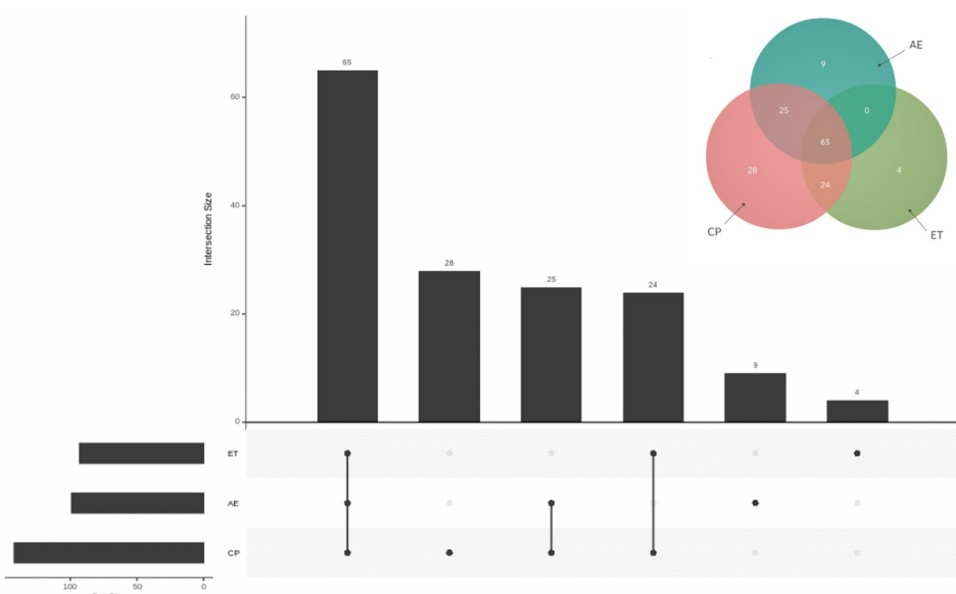

**Fig 4. Venn diagram and UpSet plot of the Argentinian samples.** Venn diagram and UpSet plot show the number of shared OTUs between chicken cecal Argentinian samples designated as ET: Experimental Trial, CP: Conventional Poultry, and AE: Agroecological Farm.

**Table 5. List of shared OTUs among chicken cecal Argentine samples.**

| Phylum | Order | Family | Genus | ET | AE | CP |
|---|---|---|---|---|---|---|
| Bacteroidetes | Bacteroidales | Bacteroidaceae | Bacteroides | 33.47 | 18.28 | 18.56 |
| Bacteroidetes | Bacteroidales | Rikenellaceae | Unclassified | 5.86 | 1.46 | 15.45 |
| Bacteroidetes | Bacteroidales | [Barnesiellaceae] | Unclassified | 5.50 | 1.27 | 8.39 |
| Firmicutes | Lactobacillales | Lactobacillaceae | Lactobacillus | 2.87 | 0.24 | 3.80 |
| Firmicutes | Clostridiales | Unclassified | Unclassified | 18.76 | 7.86 | 11.97 |
| Firmicutes | Clostridiales | Lachnospiraceae | Unclassified | 4.38 | 0.59 | 1.92 |
| Firmicutes | Clostridiales | Lachnospiraceae | [Ruminococcus] | 2.44 | 3.24 | 2.74 |
| Firmicutes | Clostridiales | Ruminococcaceae | Unclassified | 10.08 | 2.21 | 7.94 |
| Firmicutes | Clostridiales | Ruminococcaceae | Faecalibacterium | 0.36 | 0.77 | 3.60 |
| Firmicutes | Clostridiales | Ruminococcaceae | *Oscillospira* | 2.62 | 1.63 | 4.23 |
| Firmicutes | Clostridiales | Ruminococcaceae | *Ruminococcus* | 4.37 | 0.33 | 2.38 |
| Firmicutes | Clostridiales | Veillonellaceae | *Megamonas* | 0 | 1.12 | 1.91 |
| Firmicutes | Clostridiales | Veillonellaceae | *Phascolarctobacterium* | 2.49 | 5.77 | 2.39 |
| Proteobacteria | Burkholderiales | Alcaligenaceae | Sutterella | 0.97 | 2.34 | 0.64 |
| Proteobacteria | Enterobacteriales | Enterobacteriaceae | Unclassified | 1.91 | 0.84 | 0.31 |

The taxonomic classification of the shared OTUs with relative abundance above 1% is shown down to the genus level.

ET: Experimental Trial, CP: Conventional Poultry, AE: Agroecological Farm

samples from northern countries (Germany) than in samples from southern countries (Slovenia, Croatia, and Hungary). The opposite geographic gradient pattern came out for Bacteroidetes phylum (Fig 2 and S2 Table).

An analysis of the influence of geography upon microbiota of birds with fermenting crop confirmed that intra-population distances were smaller than between populations, where the differences in the crop microbiota were mostly assigned to environmental differences [6]. In another study, Hird *et al.* [7] suggested that genetics might play less influence in comparison to non-genetic factors like locality, age, and diet in shaping passerine gut microbiota. Finally, in a broader avian study, Waite and Taylor [25] attributed the composition of GIT microbiota mostly to host and location.

One of the main challenges when extracting information from public sequence resources is to take into account the biases and limitations of the methodological approaches that could confound the outcome from the bioinformatics workflow. Sequencing depth indirectly determines the abundance of the bacterial species. Indeed, the detection of rare OTUs requires the presence of many sequences per sample [26]. This constraint can be overcome by performing a normalization data step through random subsampling and total sum-scaling method. Additionally, we considered three factors related to methodological approaches (DNA extraction kit, the variable region of the 16S rRNA, and the sequencing platform) in the analysis. Although Fouhy *et al.* [27] found that the extraction method had a low effect on overall composition, Kennedy *et al.* [28] obtained significant differences in relative abundance associated with different DNA extraction methods. Walker *et al.* [29] and Fouhy *et al.* [27] found the PCR primer sequences are critical determinants of the final bacterial sequences profile. To overcome for potential bias due to both variable target region and primer pairs we used the closed reference OTU picking approach for analyzing the worldwide samples, at the risk of discarding novel real reads. Finally, Allali *et al.* [30] demonstrated significant differences between sequencing platforms and library preparation protocols in the determination of microbial diversity and species richness.

In our work, we corroborated, according to R2 values, that the sequencing platform, extraction kit, and the variable sequenced region indeed have an impact though lesser than the

geographic origin (Table 2). Moreover, the results from the two datasets from Argentina (Table 1 and Fig 1) significantly grouped together, yet coming not only from different trials but also obtained in different time-frames.

The Mantel test [31] is a powerful tool for analyzing multivariate data, particularly for data sets expressed through pairwise distances [32]. In this regard, our results showed a significant correlation between beta diversity (unweighted UniFrac) and haversine geographical distances, reinforcing the major influence of geography over any other factor that may arise from methodological constraints.

Others factors, like intestinal portion and age, seemed not impact on microbiota modulation to the same extent as geography. For instance, Malaysia samples clustered together even though they belonged to different sections of GIT (ileum and cecum) and two different age groups (21 and 42 days). On the other hand, the diet, which is one of the most studied parameter in the bibliography, could be a determining influence factor. However, some of the datasets used in our analysis included more than one treatment, but their distribution on PCoA did not show any response to this characteristic. For example Germany dataset include two groups fed with commercial or monocalcium phosphate additive, and the same happened with USA and Argentine samples, which comprised commercial diet versus organic acids, and tannins or bacitracin, respectively (Table 1).

We coined the concept of "local microbiota" because of the highly divergent poultry microbiota linked with geographical location. Resuming the perspective of human-environment geography, local microbiota mirrors the autochthonous non-genetic drivers that modulate bacterial composition. In other words, we propose that geographical location is a convergent feature that conflates non-genetic factors. Thus, local microbiota is worth to take into consideration as the proper base-line for identifying the correlation of the poultry lifestyle and the GIT microbiota, for testing additives on diet to modulate the microbiota as growth promoter factors, and for improving production performance. As an example, the in depth analysis of GIT microbiota of poultry from Argentina allowed us to characterized a native one, bearing Veillonellaceae family in a more predominantly way (> 3%) than in any of the other of the analyzed countries (below 1%) (S2 Table). Particularly, the members of this family are morphologically diverse and obligate Gram-negative anaerobes, capable of degrading organic acids, fermenting lactate, and forming intergeneric coaggregates with other bacteria providing nutrients and protection for all participants [33].

Yet we can talk about an Argentine microbiota, we still can distinguish the existence of specific community structures linked to CF versus ET at the sub local level (Fig 3 and S3 Table). Additionally, diversity indexes from CF are higher than the ones from ET (Observed OTUs, Chao1, Shannon, and Simpson indexes—Table 4). We consider the litter management regimen as the most noticeable feature among the variables that could be responsible of the observed differences between both husbandry practices. The poultry litter consists primarily of a mixture of bedding materials and bird excreta and, unlike in ET, in commercial farms the litter is used throughout the year (from five to six productive cycles). Repeated use of poultry litter, results in considerable changes in the chemical and microbiological conditions of litter [34]. Other authors had also demonstrated, the litter effect on the composition and structure of poultry GIT microbiota. In that sense, our results (Table 4), are in accordance with Wang et al. [34] and Cressman et al. [35], where the diversity of cecal samples from animals raised in pens with reused litter was significantly greater in comparison to the diversity of those raised using fresh litter.

Clearly differences were observed according to the multidimensional scaling analysis. Fig 3 displays not only the split into experimental or commercial conditions (ET and CF) but also the sub divisions within each group. ET comprises two slightly different experimental designs

(ET1 and ET2, Table 1) and CF involves two alternative productive management systems (CP and AE). Notably, although CP encompasses samples from different commercial farms (S1 Table) they are all grouped together, significantly separated from the AE ones.

Despite these differences found into Argentine samples, we described a microbial taxonomic core (Fig 4 and Table 5). The identification of a taxonomic core among Argentine poultry could be useful to evaluate the trends of microbiota dynamics in a more accurate way at regional level. The four more abundant shared OTUs were *Bacteroides*, Rikenellaceae, Clostridiales and Ruminococcacea, typical members of the chicken GIT.

The cecum harbors a bacterial community that allows anaerobic fermentation of cellulose and other substrates [19]; many of the members of this community belong to the Bacteroidetes phylum. Among Bacteroidetes, *Bacteroides* was the most abundant genus in the Argentine core, capable of performing an efficient polysaccharide degradation and producing short-chain volatile fatty acids [36]. On the other hand, the Rikenellaceae family generally indicates gastrointestinal good–health. Members of this family seem to be specialized in the digestive tract of a number of different animals, and have been identified both in fecal and GIT samples [37].

Within Firmicutes phylum, Clostridia class dominated the ileum and cecum microbiota of healthy chickens [14]. Rinttilä and Apajalahti [38] suggest that most members of Clostridia are nonpathogenic, encompassing many beneficial bacteria like cellulose and starch degraders. In accordance, the Clostridiales order was the most abundant member of the Firmicutes phylum in the Argentine core. Among the Clostridia class Ruminococcaceae and Lachnospiraceae were the most abundant families similar to what has been described by Oakley *et al*. [16] and Neumann and Suen [39] for cecum of broiler chicken.

## Conclusion

The results of the present study reinforce the role of geographic location as a native modulator factor of microbial community present in chicken gastrointestinal tract. We believe that a global picture of diversity is emerging, despite the limitations of cross-study meta-analyses due mainly to methodological biases. Therefore, here we report a conservative approach, using closed reference OTU picking due to different 16S gene regions involved, limiting the potential for sequencing noise to interfere with the results at the cost of perhaps discarding real, novel reads.

A larger number of studies should be included in future analyzes to validate the results to a wider extent, to support the similarities in the composition within the same country (or same latitude).

Additionally, this study is the first report describing the Argentine microbiota for experimental and commercial farms that could be considered as first baseline approximation when testing modulators of the GIT microbiota in specific contexts to improve poultry health and production. However, further investigations are required to better link the environmental conditions with microbiota modulation parameters so as to develop novel strategies for improving production outputs.

## Supporting information

**S1 Fig. Clustering analysis of chicken GIT microbial communities at the genus level, based on Bray-Curtis dissimilarity, from different geographic locations.**
(TIF)

**S2 Fig. Relative abundance of bacteria at the family level in the chicken GIT samples evaluated from different geographic locations.** Geographic locations were designated as AUS: Australia, ARG: Argentina, CRO: Croatia, GER: Germany, HUN: Hungary, MAL: Malaysia,

SLO: Slovenia, and USA: United States. The taxonomic classification is expressed as p_: phylum, c_: class, o_: order, and f_: family.
(TIF)

**S1 Table. Sampling data from Argentinian poultry commercial farms.**
(XLSX)

**S2 Table. Statistical analysis for relative abundance of the predominant families in the chicken GIT samples.** Chicken GIT samples were evaluated from different geographic locations designated as AUS: Australia, ARG: Argentina. CRO: Croatia, GER: Germany, HUN: Hungary, MAL: Malaysia, SLO: Slovenia, and USA: United States.
(XLSX)

**S3 Table. Statistical analysis for predominant families in chicken cecal Argentinian microbiota.** Argentinian samples were designated as CP: Conventional Poultry, ET: Experimental Trial, and AE: Agroecological Farm.
(XLSX)

## Acknowledgments

This work used computational resources from the Bioinformatics Unit, IABiMo (CICVyA-INTA/CONICET), part of the Consorcio Argentino de Tecnología Genómica (CATG) (PPL Genómica, MINCyT).

## Author Contributions

**Conceptualization:** Natalia Pin Viso, Mariano Fernández Miyakawa, Marisa Diana Farber.

**Data curation:** Natalia Pin Viso.

**Formal analysis:** Natalia Pin Viso.

**Funding acquisition:** Mariano Fernández Miyakawa, Marisa Diana Farber.

**Investigation:** Natalia Pin Viso, Mariano Fernández Miyakawa, Marisa Diana Farber.

**Methodology:** Natalia Pin Viso, Marisa Diana Farber.

**Project administration:** Mariano Fernández Miyakawa, Marisa Diana Farber.

**Resources:** Natalia Pin Viso, Enzo Redondo, Juan María Díaz Carrasco, Leandro Redondo.

**Software:** Natalia Pin Viso.

**Supervision:** Marisa Diana Farber.

**Validation:** Marisa Diana Farber.

**Visualization:** Natalia Pin Viso, Julia Sabio y. Garcia.

**Writing – original draft:** Natalia Pin Viso, Marisa Diana Farber.

**Writing – review & editing:** Natalia Pin Viso, Julia Sabio y. Garcia, Mariano Fernández Miyakawa, Marisa Diana Farber.

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
