## [Decision Letter · Decision Letter 0]

14 May 2020

PONE-D-20-04279

Geography as non-genetic modulation factor of chicken cecal microbiota

PLOS ONE

Dear Dr Farber,

Thank you for submitting your manuscript to PLOS ONE. After careful consideration, we feel that it has merit but does not fully meet PLOS ONE’s publication criteria as it currently stands. Therefore, we invite you to submit a revised version of the manuscript that addresses the points raised during the review process.

Both of the invited reviewers share concerns about the conclusions of the geographical role. I think they have valid claims, and you should carefully address their inquiries. I need to evaluate the next submission to consider it for publication. Please upload all of your data to the proper repositories, tone down the conclusions if required. 

We would appreciate receiving your revised manuscript by Jun 28 2020 11:59PM. To enhance the reproducibility of your results, we recommend that if applicable you deposit your laboratory protocols in protocols.io, where a protocol can be assigned its own identifier (DOI) such that it can be cited independently in the future. For instructions see: http://journals.plos.org/plosone/s/submission-guidelines#loc-laboratory-protocols

We look forward to receiving your revised manuscript.

Kind regards, saludos hasta Argentina!

Luis David Alcaraz, Ph.D.

Academic Editor

PLOS ONE

Journal Requirements:

2. We note that you are reporting an analysis of a microarray, next-generation sequencing, or deep sequencing data set. PLOS requires that authors comply with field-specific standards for preparation, recording, and deposition of data in repositories appropriate to their field. Please upload these data to a stable, public repository (such as ArrayExpress, Gene Expression Omnibus (GEO), DNA Data Bank of Japan (DDBJ), NCBI GenBank, NCBI Sequence Read Archive, or EMBL Nucleotide Sequence Database (ENA)). In your revised cover letter, please provide the relevant accession numbers that may be used to access these data. For a full list of recommended repositories, see http://journals.plos.org/plosone/s/data-availability#loc-omics or http://journals.plos.org/plosone/s/data-availability#loc-sequencing.

Reviewers' comments:

Reviewer's Responses to Questions

**Comments to the Author**

1. Is the manuscript technically sound, and do the data support the conclusions?

Reviewer #1: No

Reviewer #2: Partly

2. Has the statistical analysis been performed appropriately and rigorously? 

Reviewer #1: No

Reviewer #2: Yes

3. Have the authors made all data underlying the findings in their manuscript fully available?

Reviewer #1: Yes

Reviewer #2: No

4. Is the manuscript presented in an intelligible fashion and written in standard English?

Reviewer #1: Yes

Reviewer #2: Yes

5. Review Comments to the Author

Reviewer #1: In this article authors performed analysis based on 16S rRNA gene amplicon sequences from cecum/fecal/ileum broiler samples available in public databases. They compared experimental trial (ET) broilers from different geographical locations. Additionally, they included Argentina broiler farms samples taken and processed by them. In this study different geographical locations are defined as different countries. Each country included only one study. They concluded that geographical location played a major influence in shaping chicken gut microbiome over any other evaluated factor based on R2 (Adonis). Authors indicated that the sequencing platform, extraction kit and the variable sequenced region have a lower impact than the geographic origin, based on R2 values.

Major comments

I consider that the general hypothesis of this article cannot be tested using the methodology chosen by the authors. Therefore, it would not be appropriate to conclude that geography is the factor that most influences bacterial composition.

Table 2 shows the R2 of each of eight fit factors that were significant (Adonis). These included geographic location, sequenced region, diet, extraction kit, intestinal portion, age, genetic line and platform. The greatest adjustment obtained was by geographic location. However, the variables being evaluated are correlated within geographic location and this effect should be controlled. R2 inflated value may be warning an overfitting of the model. It would not be appropriate to suggest that geographic location other than differences between studies, which should be considered as the combination of multiple methodological variables, is having an effect. Thus, the closeness of samples from Slovenia, Croatia and Hungary in PCoA from figure 1 may be due to the fact that they came from the same study. Moreover, Germany and the USA, that are countries in the same latitude as Slovenia, Croatia and Hungary, were separated, and they came from different studies.

The definition of a “local microbiota” between countries should only be defined as such when at least more than one study per country or per locality is carried out. Furthermore, these studies must have a certain degree of standardization of sample processing and sequencing protocols as previous studies (1,2,3,4).

References

1. Godoy-Vitorino F, Leal SJ, Díaz WA, Rosales J, Goldfarb KC, 429 García-Amado MA, et al. Differences in crop bacterial community structure between hoatzins from different geographical locations. Res Microbiol. 2012;163:211–20.

2. Hird SM, Carstens BC, Cardiff SW, Dittmann DL, Brumfield RT. Sampling locality is more detectable than taxonomy or ecology in the gut microbiota of the brood-parasitic Brown headed Cowbird (Molothrus ater). PeerJ. 2014;2:e321.

3. Videnska P, Rahman MM, Faldynova M, Babak V, Matulova ME, Prukner-Radovcic E, et al. Characterization of egg laying hen and broiler fecal microbiota in poultry farms in Croatia, Czech Republic, Hungary and Slovenia. PLoS One. 2014;9(10):e110076.

4. Zhou X, Jiang X, Yang C, Ma B, Lei C, Xu C, et al. Cecal microbiota of Tibetan Chickens from five geographic regions were determined by 16S rRNA sequencing.

Reviewer #2: The manuscript presented by Pin Viso and collaborators presents a meta-analysis of broilers' cecal microbiota from multiple countries and self-generated data about Argentinian poultry cecal microbiota. The analysis and the data presented are well processed, using standard pipelines. However, you did not declare precise run parameters, and I suggest to writhe the whole bioinformatics pipeline as a piece of supplementary information uploaded to figshare or Github. I would encourage the authors to seek alternative explanations beyond geography for microbiome structuring. The declared diets for all the other studies are broilers' standardized commercial diets (SDC), with slight modifications in each country. Previous works have shown that tiny changes in diet could alter the microbiome output in chickens like a recent publication adding galactooligosaccharides to chicken diet (https://msystems.asm.org/content/5/1/e00827-19). Another possible explanation could be poultry management in each location. Adonis results are suggestive of supporting significant geographical effects further. I would recommend using Mantel tests to correlate geographical distances among locations and the microbiome compositions. Some of your conclusions have no data back, and you could move them to the discussion. Another possibility is the sequence length of each study since you are combining multiple sequencing platforms with different efficiency and output. Please deposit OTU tables, taxonomic assignments, representative sequences in a repository (figshare) to share among other people working in bird microbiomes, and it would facilitate further meta-analysis

Particular comments:

L29 Missing comma "genotype, or geographical"

L93-94 Table 1. Please upload your data to NCBI/EBI/DDBJ (AVAILABLE UNDE REQUEST). Add sequence length average of each platform, and the total output in base pairs for each compared project.

L118-119 Add details about PCR experiments run conditions, primers used, etc.

L128-145 Description is neat, however for using reproducibility, please prepare a notebook as supplementary information; you could use the aid of jupyter-notebooks or merely a text with all your bioinformatics and statistical procedures cut-offs.

L150 What is the alpha cut-off value of STAMP?

Table 2 Please include average sequence length and sequencing output for the multiple datasets in your analysis using ADONIS.

L185 Missing comma, check out the use of commas with lists in English: https://www.grammarly.com/blog/comma/ not quite the same as in Spanish with the last conjunction.

L208 Rewrite from "under ET" to "under experimental trail (ET)"

L214 How many OTUs did you find in all the compared samples? The overall alpha diversity metrics.

Since space is no limitation in PLoS one, I would recommend you upload Supplemental Figure 1 to the main text to have the microbial structure description of your comparisons.

Table 4 there are some inconsistencies between the Shannon values reported here and the ones reported in Table 3 for Argentina, please clarify.

L285 Rewrite "more prevelailing" to "relevant"

L288 "infant", please describe human infants.

L289 Missing comma "Enterobacteriaceae, and"

L310 Define TSS method

L326 Discuss poultry management, diet, etc.

L333 Define what were the experimental treatments in the other works

L338 "local microbiota" to endemic?

L343 and/or Please choose one

L374 Describing a microbial taxonomic core is always possible it's just set theory. So rewrite to "we described a microbial taxonomic core"

L374 I recommend you to try Upset to calculate your core and even compare with the other described works. https://cran.r-project.org/web/packages/UpSetR/vignettes/basic.usage.html

L374-375 Speculation, what is the basis for stability or homeostasis. How do you measure it?

L377 remove prefixes g__ f__ c__

L389 ellaborate the "All existing evidence"

L400-402 Move this to the discussion, where you can elaborate on your idea. However, it is outside the scope of your manuscript. "This local microbiota could improve the understanding of the poultry lifestyle on production performance"

L404 modify to "first baseline approximation"

L405 rewrite "researches" to "investigations"

6. PLOS authors have the option to publish the peer review history of their article (what does this mean?). If published, this will include your full peer review and any attached files.

Reviewer #1: No

Reviewer #2: No

---

## [Author Response · Author response to Decision Letter 0]

6 Jul 2020

PONE-D-20-04279

Geography as non-genetic modulation factor of chicken cecal microbiota

We are sending a revised version of our manuscript. We are grateful to the Editor and the Reviewers for the attentive reading of our manuscript and helpful insight. The changes suggested by the Reviewers have contributed substantially to improving the manuscript. We carefully evaluated the suggestions, revised the text and hope the manuscript is acceptable for the publication in the present format. 

Our responses (in italics) to Reviewers' comments:

Review Comments to the Author

Reviewer #1: In this article authors performed analysis based on 16S rRNA gene amplicon sequences from cecum/fecal/ileum broiler samples available in public databases. They compared experimental trial (ET) broilers from different geographical locations. Additionally, they included Argentina broiler farms samples taken and processed by them. In this study different geographical locations are defined as different countries. Each country included only one study. They concluded that geographical location played a major influence in shaping chicken gut microbiome over any other evaluated factor based on R2 (Adonis). Authors indicated that the sequencing platform, extraction kit and the variable sequenced region have a lower impact than the geographic origin, based on R2 values.

Major comments

I consider that the general hypothesis of this article cannot be tested using the methodology chosen by the authors. Therefore, it would not be appropriate to conclude that geography is the factor that most influences bacterial composition.

Table 2 shows the R2 of each of eight fit factors that were significant (Adonis). These included geographic location, sequenced region, diet, extraction kit, intestinal portion, age, genetic line and platform. The greatest adjustment obtained was by geographic location. However, the variables being evaluated are correlated within geographic location and this effect should be controlled. R2 inflated value may be warning an overfitting of the model. It would not be appropriate to suggest that geographic location other than differences between studies, which should be considered as the combination of multiple methodological variables, is having an effect. Thus, the closeness of samples from Slovenia, Croatia and Hungary in PCoA from figure 1 may be due to the fact that they came from the same study. Moreover, Germany and the USA, that are countries in the same latitude as Slovenia, Croatia and Hungary, were separated, and they came from different studies.

The definition of a “local microbiota” between countries should only be defined as such when at least more than one study per country or per locality is carried out. Furthermore, these studies must have a certain degree of standardization of sample processing and sequencing protocols as previous studies (1,2,3,4).

References

1. Godoy-Vitorino F, Leal SJ, Díaz WA, Rosales J, Goldfarb KC, 429 García-Amado MA, et al. Differences in crop bacterial community structure between hoatzins from different geographical locations. Res Microbiol. 2012;163:211–20.

2. Hird SM, Carstens BC, Cardiff SW, Dittmann DL, Brumfield RT. Sampling locality is more detectable than taxonomy or ecology in the gut microbiota of the brood-parasitic Brown headed Cowbird (Molothrus ater). PeerJ. 2014;2:e321.

3. Videnska P, Rahman MM, Faldynova M, Babak V, Matulova ME, Prukner-Radovcic E, et al. Characterization of egg laying hen and broiler fecal microbiota in poultry farms in Croatia, Czech Republic, Hungary and Slovenia. PLoS One. 2014;9(10):e110076.

4. Zhou X, Jiang X, Yang C, Ma B, Lei C, Xu C, et al. Cecal microbiota of Tibetan Chickens from five geographic regions were determined by 16S rRNA sequencing.

We agree with the Reviewer that we ought to control for any over-fitting of the model. However, given the scarcity of more than one data set per country we alternatively selected pairs of experiments with the same state for any variable, so as to support the comparison. That is, the data set from USA and Australia target the 16S V1-V3 region, whereas V3-V4 region was the one used in the data sets from Eastern Europe and Argentina. Furthermore, Illumina was the sequencing platform used in the experiments from Argentina and Malaysia, and Roche for Eastern Europe, Germany and the USA ones. Additionally, we performed the Mantel test obtaining a significant correlation between distances matrix from geographic and microbial composition. 

Since public datasets were used, the sampling and sequencing process cannot be standardized ex-post. However, we still think that valid information could be deduced from cumulative data available in public resources. For stripping out the methodology effect we analyzed as much variables as possible, such as the DNA extraction kit, the sequencing primers or amplified region, and the sequencing platform. Although they exerted significant effect, their influence was lesser than the one coming from the location (table 2). Finally, we coined the “local microbiota” definition after testing the location hypothesis using Argentinian data coming from different trials and time-frames. For this analysis we fulfilled the suggestion of using at least more than one set per locality. 

Reviewer #2: The manuscript presented by Pin Viso and collaborators presents a meta-analysis of broilers' cecal microbiota from multiple countries and self-generated data about Argentinian poultry cecal microbiota. The analysis and the data presented are well processed, using standard pipelines. However, you did not declare precise run parameters, and I suggest to writhe the whole bioinformatics pipeline as a piece of supplementary information uploaded to figshare or Github. I would encourage the authors to seek alternative explanations beyond geography for microbiome structuring. The declared diets for all the other studies are broilers' standardized commercial diets (SDC), with slight modifications in each country. Previous works have shown that tiny changes in diet could alter the microbiome output in chickens like a recent publication adding galactooligosaccharides to chicken diet (https://msystems.asm.org/content/5/1/e00827-19). Another possible explanation could be poultry management in each location. Adonis results are suggestive of supporting significant geographical effects further. I would recommend using Mantel tests to correlate geographical distances among locations and the microbiome compositions. Some of your conclusions have no data back, and you could move them to the discussion. Another possibility is the sequence length of each study since you are combining multiple sequencing platforms with different efficiency and output. Please deposit OTU tables, taxonomic assignments, representative sequences in a repository (figshare) to share among other people working in bird microbiomes, and it would facilitate further meta-analysis

We appreciate the suggestions regarding seeking for alternative explanations. Although we sustained the geographical location as the driver we went deeper into the subject of Geography definitions for gaining clarity. To that end, we supported our argument using specific bibliography (Turner 2002, Siso Quintero 2010) and recalling the human-environment theoretical position, we proposed that geographical location was a convergent feature that conflated non-genetic factors (added in “Introduction” L75-81and “Discussion” L359-363.).

On the other hand, we acknowledge the suggestions of using Mantel test to support the results (L181-183). In addition, QIIME scripts together with intermediate results can be reach trough figshare: https://doi.org/10.6084/m9.figshare.c.4993856

Particular comments:

L29 Missing comma "genotype, or geographical"

Correction has been done

L93-94 Table 1. Please upload your data to NCBI/EBI/DDBJ (AVAILABLE UNDE REQUEST). Add sequence length average of each platform, and the total output in base pairs for each compared project.

All sequence data generated for this paper was deposited in the NCBI Sequence Read Archive database under the BioProject accession number PRJNA579062. Additionally, QIIME scripts and intermediate results are available from figshare (https://doi.org/10.6084/m9.figshare.c.4993856).

Sequenced cited in Table 1 were obtained after requesting to J. Diaz Carrasco (Diaz Carrasco et al. 2018 and Diaz Carrasco unpublished).

We consider unnecessary to add sequence length average of each platform in Table 1, as long as the resulting files, after merging Forward (F) and Reverse (R) ones, turn out to be the 16S amplified region which range of variation is already considered in Table 1 under “Region sequenced”. Similarly, data is subjected to normalization before composition and alpha and beta diversity analysis, so total output won’t exert any influence. 

L118-119 Add details about PCR experiments run conditions, primers used, etc. 

Thank you for bringing up to this matter. Not only sequencing of the samples but also amplification were performed at Macrogen Inc,(we reworded that paragraph). Nucleotide primer sequences were added in L131--132.

L128-145 Description is neat, however for using reproducibility, please prepare a notebook as supplementary information; you could use the aid of jupyter-notebooks or merely a text with all your bioinformatics and statistical procedures cut-offs.

Added in L153-154. QIIME scripts and intermediate results are available from figshare (https://doi.org/10.6084/m9.figshare.c.4993856) 

L150 What is the alpha cut-off value of STAMP?

Added in L171

Table 2 Please include average sequence length and sequencing output for the multiple datasets in your analysis using ADONIS. 

Answered above, in relation to Table 1 suggestions. 

L185 Missing comma, check out the use of commas with lists in English: https://www.grammarly.com/blog/comma/ not quite the same as in Spanish with the last conjunction.

Corrections have been done all along the text.

L208 Rewrite from "under ET" to "under experimental trail (ET)"

Modification has been done (L223)

L214 How many OTUs did you find in all the compared samples? The overall alpha diversity metrics. 

Added (L228).

Since space is no limitation in PLoS one, I would recommend you upload Supplemental Figure 1 to the main text to have the microbial structure description of your comparisons.

We decline this suggestion, we consider Fig 2 depicts the microbial structure satisfactorily. It is our understanding that Supp Fig 1 would not add fundamental information that worth to include in the main text. 

Table 4 there are some inconsistencies between the Shannon values reported here and the ones reported in Table 3 for Argentina, please clarify. 

The data set analyzed in Table 4 is rather different from the one analyzed in Table 3. For the Characterization of Argentine cecal microbiota (Table 4) we consider not only the Experimental Treatment data used in the among countries comparison (ARG_ET 1 & ARG_ET2, Table 1), but also 17 extra samples that came from the same data set but were from a different animal category (wider range of ages). Reworded in L226. 

L285 Rewrite "more prevelailing" to "relevant"

Modification has been done (L302)

L288 "infant", please describe human infants.

Infant category corresponded to 6 weeks of age (Added in L307)

L289 Missing comma "Enterobacteriaceae, and"

Done

L310 Define TSS method

Added in L329, described in MyM L145-146

L326 Discuss poultry management, diet, etc.

Factors such as intestinal portion, age, and diet are discussed from L349 to L359. Additionally, we added more precise information in the paragraph corresponding to “local microbiota” definition (L360-364).

L333 Define what were the experimental treatments in the other works

Treatments are defined in Table 1 (Diet column) and all along the paragraph from L349 to L359

L338 "local microbiota" to endemic?

It is our understanding that endemic species are found only in the area under consideration and nowhere else. GIT is a rather stable environment in itself, and the variation among microbiota is mainly due to changes in the relative abundance. Our definition of “local microbiota” is in the framework of human-environment geography, being location an idealized geographical space.

L343 and/or Please choose one

Modification has been done (L 367)

L374 Describing a microbial taxonomic core is always possible it's just set theory. So rewrite to "we described a microbial taxonomic core"

Modification has been done ( L 397-398)

L374 I recommend you to try Upset to calculate your core and even compare with the other described works. https://cran.r-project.org/web/packages/UpSetR/vignettes/basic.usage.html

We acknowledge the suggestion. We modified the figure 4 for improving further interpretations. We are uploading it as “new_Fig4”

L374-375 Speculation, what is the basis for stability or homeostasis. How do you measure it?

We agree is mere speculation, being out of the reach of our results at this stage. We deleted the phrase L398-399

L377 remove prefixes g__ f__ c__

Done L402

L389 ellaborate the "All existing evidence"

We added the reference of Rinttilä and Apajalahti, a revision work that thoroughly described microbiota implications for broiler chickens health L413

L400-402 Move this to the discussion, where you can elaborate on your idea. However, it is outside the scope of your manuscript. "This local microbiota could improve the understanding of the poultry lifestyle on production performance"

We agree is just speculation, we deleted the phrase in accordance to the suggestion of toning down the conclusions. L425-426

L404 modify to "first baseline approximation"

Done ( L427-428)

L405 rewrite "researches" to "investigations"

Done ( L430)

---

## [Decision Letter · Decision Letter 1]

8 Oct 2020

PONE-D-20-04279R1

Geography as non-genetic modulation factor of chicken cecal microbiota

PLOS ONE

Dear Dr. Farber,

Thank you for submitting your manuscript to PLOS ONE. After careful consideration, we feel that it has merit but does not fully meet PLOS ONE’s publication criteria as it currently stands. Therefore, we invite you to submit a revised version of the manuscript that addresses the points raised during the review process.

Please address comments of the reviewers, particularly reviewer 1 and 4 and consider toning down of your conclusions considering their comments. And I will consider acceptance. 

We look forward to receiving your revised manuscript.

Kind regards,

Luis David Alcaraz, Ph.D.

Academic Editor

PLOS ONE

Reviewers' comments:

Reviewer's Responses to Questions

**Comments to the Author**

1. If the authors have adequately addressed your comments raised in a previous round of review and you feel that this manuscript is now acceptable for publication, you may indicate that here to bypass the “Comments to the Author” section, enter your conflict of interest statement in the “Confidential to Editor” section, and submit your "Accept" recommendation.

Reviewer #1: All comments have been addressed

Reviewer #3: (No Response)

Reviewer #4: All comments have been addressed

2. Is the manuscript technically sound, and do the data support the conclusions?

Reviewer #1: No

Reviewer #3: Yes

Reviewer #4: Partly

3. Has the statistical analysis been performed appropriately and rigorously? 

Reviewer #1: No

Reviewer #3: Yes

Reviewer #4: Yes

4. Have the authors made all data underlying the findings in their manuscript fully available?

Reviewer #1: Yes

Reviewer #3: No

Reviewer #4: Yes

5. Is the manuscript presented in an intelligible fashion and written in standard English?

Reviewer #1: Yes

Reviewer #3: Yes

Reviewer #4: Yes

6. Review Comments to the Author

Reviewer #1: I am concerned about the over fitting the authors can not control due to the scarcity of more than one data set per country, particularly important because the whole article is focused in a conclusion that can not be well supported. Mantel analysis including geographical distance may be also bias do to the presence of a single study which includes 3 different countries that are geographically close to each other. I consider it essential to detect similarities in composition among different studies from the same country (or same latitude) to suggest that geography, more than studies, is the main compositional shaping variable.

Reviewer #3: The authors present a meta-analysis of cecum/ileum and fecal broiler microbiota based on 16S rRNA gene amplicon sequences taken from public databases including studies in different countries and compared them with their experimental data from broiler samples with different management practices in Argentina. Their conclusions reinforce notions of the geographic location as driver of microbial community composition since other tested factors are not informative enough. Data analysis is sufficiently well done using standard pipelines. Reviewers statistical and technical inquiries have been addressed, as well as the elimination of non supported conclusions.

Detailed descriptions of bioinformatic methods and other inquiries made by Reviewer #2 cannot be consulted. The figshare link is not available at https://doi.org/10.6084/m9.figshare.c.4993856

Particular comments:

L204 . Lactoballaceae rewrite Lactobacillaceae

L366 Rewrite non-genetic

L369 Rewrite local microbiota is worth

L372 Rewrite in depth

L419 Rewrite plural for degrader

Reviewer #4: I found that this is a second round review and the authors already responded to the the previous reviewers comments. I agree with all concerns raised by the previous reviewers. The authors performed the study and analyzed data carefully considering many potential pitfalls that are inherent to this type of meta analysis. The authors attempted to address all the critiques of the previous reviewers. Nonetheless, I think the validity of the conclusion in this study that geography is a major variable for chicken gut microbiota cannot be supported fully with no room for further discussion. Therefore, the conclusion of this study still has limitations, yet the data presented here has certain value to the research community. I came to the conclusion to support the publication of this manuscript under the following conditions.

1. The authors add a paragraph to present inherent limitations of this study and potential issues associated with the conclusion of the study (as well pointed out by the reviewers 1 and 2 from the 1st round review).

2. I ask the authors to add the information on the PCR primers used in different studies (probably in a supplementary table). Even though two different studies used the same V3-V4 region (for example) they could have used different primer designs (primer length, precise targeting region, degeneracy etc.). Since the variation in the primer design can bring significant changes in the resulting 16S rRNA gene profiles, it is necessary to include the information.

7. PLOS authors have the option to publish the peer review history of their article (what does this mean?). If published, this will include your full peer review and any attached files.

Reviewer #1: No

Reviewer #3: No

Reviewer #4: No

---

## [Author Response · Author response to Decision Letter 1]

13 Oct 2020

Reviewer #1: I am concerned about the over fitting the authors can not control due to the scarcity of more than one data set per country, particularly important because the whole article is focused in a conclusion that cannot be well supported. Mantel analysis including geographical distance may be also bias do to the presence of a single study which includes 3 different countries that are geographically close to each other. I consider it essential to detect similarities in composition among different studies from the same country (or same latitude) to suggest that geography, more than studies, is the main compositional shaping variable.

As we mentioned in the first revision, we agree with the Reviewers that we cannot totally control the over-fitting of the model. In an effort to track for any trend in a more stringent manner, we compared pairs of experiments with the same state for any variable, so as to support the comparison (at least two data sets targeting the same 16S region, at least two that had been used the same sequencing platform, etc.). Additionally, we used two data sets from Argentina coming from different trials and time-frames. 

We re-analyzed all data set from geographical location analysis, using a more conservative close reference approach (Line 138-141 M&M), obtaining equivalent results, revealing the geographical trend that prompted our work (Results: Fig1, Table 2, Fig, 2, Table 3, Fig S2, Table S2).

We have rewritten the conclusion section to account for inherent limitations of this kind of studies. 

Reviewer #3: The authors present a meta-analysis of cecum/ileum and fecal broiler microbiota based on 16S rRNA gene amplicon sequences taken from public databases including studies in different countries and compared them with their experimental data from broiler samples with different management practices in Argentina. Their conclusions reinforce notions of the geographic location as driver of microbial community composition since other tested factors are not informative enough. Data analysis is sufficiently well done using standard pipelines. Reviewers statistical and technical inquiries have been addressed, as well as the elimination of non supported conclusions.

Detailed descriptions of bioinformatic methods and other inquiries made by Reviewer #2 cannot be consulted. The figshare link is not available at https://doi.org/10.6084/m9.figshare.c.4993856

We have made publicly available the additional information under the link: https://doi.org/10.6084/m9.figshare.c.4993856.v1

Particular comments:

L204 . Lactoballaceae rewrite Lactobacillaceae 

Has been corrected in L 196

L366 Rewrite non-genetic

Has been corrected in L 366

L369 Rewrite local microbiota is worth

Has been corrected in L 369

L372 Rewrite in depth

Has been corrected in L 372

L419 Rewrite plural for degrader

Has been corrected in L 418

Reviewer #4: I found that this is a second round review and the authors already responded to the the previous reviewers comments. I agree with all concerns raised by the previous reviewers. The authors performed the study and analyzed data carefully considering many potential pitfalls that are inherent to this type of meta-analysis. The authors attempted to address all the critiques of the previous reviewers. Nonetheless, I think the validity of the conclusion in this study that geography is a major variable for chicken gut microbiota cannot be supported fully with no room for further discussion. Therefore, the conclusion of this study still has limitations, yet the data presented here has certain value to the research community. I came to the conclusion to support the publication of this manuscript under the following conditions.

1. The authors add a paragraph to present inherent limitations of this study and potential issues associated with the conclusion of the study (as well pointed out by the reviewers 1 and 2 from the 1st round review).

We have rewritten and toned down the conclusion accordingly with the suggestion. 

2. I ask the authors to add the information on the PCR primers used in different studies (probably in a supplementary table). Even though two different studies used the same V3-V4 region (for example) they could have used different primer designs (primer length, precise targeting region, degeneracy etc.). Since the variation in the primer design can bring significant changes in the resulting 16S rRNA gene profiles, it is necessary to include the information.

In an recent work (Mancabelli L et al., Microorganisms 2020, 8, 131; doi:10.3390/microorganisms801013), the authors designed a study in order to identify possible correlations between bacterial amplification capabilities and the PCR

primer pairs. To that end, they considered different combinations of primers per each variable region (for example 6 primer pairs targeting the V3-V4), demonstrating that the detected community structure strictly depend on the

particular hypervariable region that was targeted for amplification. Nevertheless, we re-analyzed all geographical data using close reference OTUs picking, as a more conservative approach We have rewritten M&M, Results and discussion session, reporting the new values that account for equivalent geographical trend depicted in the previous version.

---

## [Decision Letter · Decision Letter 2]

16 Dec 2020

Geography as non-genetic modulation factor of chicken cecal microbiota

PONE-D-20-04279R2

Dear Dr. Farber,

We’re pleased to inform you that your manuscript has been judged scientifically suitable for publication and will be formally accepted for publication once it meets all outstanding technical requirements.

Kind regards,

Luis David Alcaraz, Ph.D.

Academic Editor

PLOS ONE

Additional Editor Comments (optional):

I detected a minor typo in L415, replace OUT to OTU

Reviewers' comments:

Reviewer's Responses to Questions

**Comments to the Author**

1. If the authors have adequately addressed your comments raised in a previous round of review and you feel that this manuscript is now acceptable for publication, you may indicate that here to bypass the “Comments to the Author” section, enter your conflict of interest statement in the “Confidential to Editor” section, and submit your "Accept" recommendation.

Reviewer #3: All comments have been addressed

2. Is the manuscript technically sound, and do the data support the conclusions?

Reviewer #3: (No Response)

3. Has the statistical analysis been performed appropriately and rigorously? 

Reviewer #3: (No Response)

4. Have the authors made all data underlying the findings in their manuscript fully available?

Reviewer #3: (No Response)

5. Is the manuscript presented in an intelligible fashion and written in standard English?

Reviewer #3: (No Response)

6. Review Comments to the Author

Reviewer #3: (No Response)

7. PLOS authors have the option to publish the peer review history of their article (what does this mean?). If published, this will include your full peer review and any attached files.

Reviewer #3: No

---

## [Editor Report · Acceptance letter]

21 Dec 2020

PONE-D-20-04279R2 

Geography as non-genetic modulation factor of chicken cecal microbiota 

Dear Dr. Farber:

I'm pleased to inform you that your manuscript has been deemed suitable for publication in PLOS ONE. Congratulations! Your manuscript is now with our production department. 

Kind regards, 

on behalf of

Dr. Luis David Alcaraz 

Academic Editor

PLOS ONE